# Detection of *Leishmania* spp. in Chronic Dermatitis: Retrospective Study in Exposed Horse Populations

**DOI:** 10.3390/pathogens11060634

**Published:** 2022-05-31

**Authors:** Alessia Libera Gazzonis, Giulia Morganti, Ilaria Porcellato, Paola Roccabianca, Giancarlo Avallone, Stefano Gavaudan, Cristina Canonico, Giulia Rigamonti, Chiara Brachelente, Fabrizia Veronesi

**Affiliations:** 1Department of Veterinary Medicine and Animal Sciences, Università degli Studi di Milano, Via dell’Università 6, 26900 Lodi, Italy; alessia.gazzonis@unimi.it (A.L.G.); paola.roccabianca@unimi.it (P.R.); 2Department of Veterinary Medicine, University of Perugia, Via San Costanzo 4, 06126 Perugia, Italy; giulia.morganti@unipg.it (G.M.); ilariaporcellatodvm@gmail.com (I.P.); giulia.rigamonti@studenti.unipg.it (G.R.); fabrizia.veronesi@unipg.it (F.V.); 3Department of Veterinary Medical Sciences, University of Bologna, Via Tolara di Sopra 50, 40064 Bologna, Italy; giancarlo.avallone@unibo.it; 4Istituto Zooprofilattico Sperimentale dell’Umbria e delle Marche, Via Varano 286, 60100 Ancona, Italy; s.gavaudan@izsum.it (S.G.); c.canonico@izsum.it (C.C.)

**Keywords:** *Leishmania*, horse, chronic dermatitis, skin biopsies

## Abstract

*Leishmania infantum* is a protozoan causing human zoonotic visceral leishmaniasis (ZVL) and visceral–cutaneous canine leishmaniosis (CanL) in the Mediterranean Basin. *L. infantum* is able to infect a large number of wild and domestic species, including cats, dogs, and horses. Since the 1990s, clinical cases of equine leishmaniasis (EL), typically characterized by cutaneous forms, have been increasingly diagnosed worldwide. The aim of the present study was to evaluate the presence of clinical forms of EL in CanL-endemic areas in Italy, where exposure of equine populations was ascertained from recent serological surveys. For this purpose, formalin-fixed and paraffin-embedded skin biopsies of 47 horses presenting chronic dermatitis compatible with EL were retrospectively selected for the study and subjected to conventional and q-PCR. A singular positivity for *L. infantum* was found; BLAST analysis of sequence amplicons revealed a 99–100% homology with *L. infantum* sequences. The histological examination revealed a nodular lymphoplasmacytic and histiocytic infiltrate; immunohistochemistry showed rare macrophages containing numerous positive amastigotes. The present retrospective study reports, for the first time, a case of a cutaneous lesion by *L. infantum* occurring in an Italian horse. Pathological and healthy skin samples should be investigated on a larger scale to provide information on the potential clinical impact of EL in the practice, and to define the role of horses in epidemiological ZVL and CanL scenarios.

## 1. Introduction

Leishmaniasis is a vector-borne disease with worldwide distribution caused by protozoan species belonging to the genus *Leishmania* (Kinetoplastida, Trypanosomatidae). Depending on the species involved, different clinical forms having a wide spectrum of clinical signs are described: cutaneous leishmaniasis (CL), visceral leishmaniasis (VL), and mucocutaneous leishmaniasis (MCL) [1].

In Europe, *Leishmania infantum*, the causative agent of zoonotic visceral leishmaniasis (ZVL) and canine leishmaniosis (CanL), is widespread in the Mediterranean Basin, including Italy, where CanL is considered endemic in almost the entire territory [2,3].

In addition to dogs, considered the main reservoir of infection, *L. infantum* has also been found in other host species, including wildlife, livestock, and other companion animals, such as cats and horses [4,5,6,7]. While the available data about the spread of the infection in the feline population have led researchers to consider the cat as a potential secondary reservoir of *L. infantum* in the endemic areas for ZVL and CanL [8], the role of the horse in the cycle of *Leishmania* spp. remains to be clarified. In fact, the benign nature of the lesions and the few clinical case reports available worldwide have led to the horse being disregarded as a potential host of *Leishmania* [7].

The first case of equine leishmaniosis (EL) was described in 1927 in Argentina [9]. Since then, cases of EL have also been described in donkeys (*Equus asinus*) and mules (*Equus caballus* × *Equus asinus*), especially in areas of Latin America (i.e., Brazil and Venezuela) endemic to human leishmaniasis and CanL [10,11,12,13].

Since the second half of the 1990s, reports of EL have progressively increased in both historically endemic but also non-endemic countries of the Old and New World [7], still representing limited numbers compared to those observed in other animal species (i.e., canids and felids).

Several *Leishmania* species may act as etiological agents of EL. In the Old World, the most frequently isolated species is *L. infantum* [14,15,16], while in the New World, the greatest number of reports are caused by *Leishmania braziliensis* [17,18], responsible for MCL. In addition, infections caused by *Leishmania martiniquensis* (formerly, *Leishmania siamensis*) [19] were detected in horses in both Europe [20] and North America [21,22,23].

Clinical cases of EL are associated, independently from the *Leishmania* species, with cutaneous forms characterized by solitary or multiple papules or nodules, with a tendency to ulcerate, mainly located on the head and limbs but also sporadically on the neck, genitals, abdomen, and thorax [17,18,24,25]. The lesions are of modest severity and with a tendency to self-resolve in a few months [7,14,26].

As the symptoms of EL may be confused with those of other diseases, diagnostic confirmation of the presence of parasites within the lesions remains mandatory. Traditionally, direct detection of parasites is accomplished with microscopic examination of specimens, consisting of cutaneous cytological smears or skin biopsies; however, examination of histological sections stained with hematoxylin and eosin (H&E) or Giemsa may be frequently inconclusive, due to the low number of amastigotes and the lack of knowledge regarding the specific inflammatory patterns.

The histopathologic lesion patterns developing in the skin of EL have not been clearly established because most studies are based on very few cases and well-defined criteria to classify the histological lesions are not available. However, the more frequent histopathological findings of EL caused by *L. infantum* [14,15], as well as those by *L. brasiliensis* [27] and *L. martiniquensis* [18,20,21,23], consist of a nodular to diffuse granulomatous inflammation composed of variable numbers of macrophages, small mature lymphocytes, multinucleated giant cells, and epithelioid cells. A slightly different histopathological aspect was described in a single study [23], reporting a moderate number of eosinophils and the absence of multinucleated giant cells in the dermal infiltrates.

In humans and dogs, several different patterns of cutaneous inflammation have been associated with susceptibility or resistance to LVZ and CanL.

To date, the pathogenetic mechanisms leading to clinical signs of EL are poorly understood and explored. However, equids are considered resistant to *Leishmania* due to the development of an effective cell-mediated immune response that allows them to control the infection, preventing progression into a systemic disease [7,28].

Recent epidemiological studies have proven that equine populations living in different regions of central and northern Italy are exposed to *Leishmania* infection [29,30], with a prevalence similar to that observed in domestic reservoir species (i.e., dogs, cats); on the other hand, to date, clinical cases of EL have never been described.

The present retrospective study aimed to evaluate the presence of *Leishmania* amastigotes in formalin-fixed and paraffin-embedded (FFPE) skin biopsies from horses living in CanL-endemic areas and having a general histopathologic diagnosis of chronic dermatitis. This will enable an assessment of the potential impact of EL in the clinical setting, the necessity (or not) of including EL in the list of differentials of chronic skin lesions, and a better definition of inflammatory patterns of EL.

## 2. Results

Lesion skin samples were obtained from 47 horses from different regions of north-central Italy and a few (n. 2) from southern Italy.

Age was available in 37 horses and ranged from 2 to 28 years (mean 11.7). In total, 20 animals were males (13 of which were neutered), and 13 were mares; the gender was not known in 14 horses. Breeds included Sella Italiano (n. 6), Quarterhorse (n. 6), English thoroughbred (n. 2), Hanoverian/Westfalen (n. 3), Trotter (n. 3), Arabian (n. 3), Shetland Pony (n. 2), Andalusian, Criollo, Haflinger, Lusitano, Lipizzaner, Holland saddle, and Polish (1 each). The breed was not known in 15 cases. Lesions were chronic, as all contained a variable number of lymphocytes, plasma cells, macrophages, and eosinophils.

One single specimen (Horse ID: 18) tested positive for both the ITS-1 and k-DNA minicircle PCRs, and also for qPCR with a parasite load of 158,000 copies. Sequencing of the ITS-1 and k-DNA minicircle amplicons produced 278 bp and 120 bp sequences, respectively, and BLAST analysis revealed a 99–100% homology with *L. infantum* sequences deposited in GenBank (FJ464412, X84238) for ITS-1, and 99–100% homology with the GenBank sequences KY950674, EU437406, and AF167712 of *L. infantum* for k-DNA minicircle.

The ITS-1 sequence obtained in the present study was submitted to GenBank under the accession number OM055847.

The skin sample that tested positive belonged to a female, 8.5-year-old Sella Italiano horse from Bologna, which showed a nodular focal lesion on the head in the temporal region, near the right ear. Histology, in this case, was characterized by expansion of the dermis by nodular dermatitis occupying the superficial, mid and deep dermis, and extending multifocally in the subcutis. The inflammation was composed of numerous histiocytes and lymphocytes, with fewer epithelioid cells and multinucleated giant cells, and lesser plasma cells. Inflammatory cells were often arranged in multinodular aggregates with central giant cells and macrophages, peripheral lymphocytes, and plasma cells. Lymphocytes multifocally formed pseudo-follicular structures, with central areas reminiscent of germinal centers. Periodic acid–Schiff, Grocott, and Ziehl–Neelsen stains were negative for fungi or acid-fast bacteria. Immunohistochemistry revealed the presence of *Leishmania* amastigotes, either as multiple or single organisms in the cytoplasm of epithelioid macrophages (Figure 1).

In the other horses included in the study, lesions ranged in terms of distribution, severity, pattern, and type of inflammatory cells. The most common patterns of inflammation were perivascular (15 cases) and interstitial (15 cases), followed by nodular (13 cases) and diffuse (4 cases). The dermatitis pattern was observed in all cases, although in 10 cases, lesions were characterized by other associated and most prominent patterns such as pustular dermatitis (2 cases), folliculitis/furunculosis (3 cases), vasculitis (4 cases), and hyperkeratosis (1 case). The formation of lymphoid follicles was detected in four cases.

In 25 cases (54%), eosinophils represented ≥20% of inflammatory cells, and lesions were classified as either eosinophilic granuloma, insect- and arthropod-bite reactions, or habronematidosis.

The details on the severity, pattern, and distribution of the inflammatory infiltrate in the study samples are represented in Appendix A.

## 3. Discussion

Exposure of equine populations from north-central areas of Italy to *L. infantum* was ascertained by recent seroepidemiological surveys showing seroprevalence values ranging from 6.4 to 13.9% [29,30], similar to those reported in dogs in the same study areas (2.5–40.3%) [31,32,33,34]; however, to date, no clinical case of EL has been described.

The present retrospective study reports, for the first time, a case of a cutaneous lesion by *L. infantum* occurring in a Sella Italiano horse from the Emilia Romagna region; since the present study is a retrospective investigation, few data were available on the infected subject, e.g., the recent and remote medical history, the clinical follow-up, and the location where the horse was housed. This information would have provided a more complete picture, both from a clinical and epidemiological point of view.

Association between positivity to *Leishmania* and risk factors such as age, gender, and breed is not well-understood [7]; however, some studies showed higher seropositivity values in horses used for recreational purposes than in farming and sporting horses, suggesting that management type may modulate the spread of *Leishmania* infection in the equine population [30,35].

The genetic identity based on the analysis of the regions within the ITS-1, ss-rRNA, and k-DNA minicircle confirmed that the case of EL was caused by *L. infantum*, the most widely spread species within the *Leishmania* genus, occurring in Italy as well as in the overall Mediterranean area.

This finding confirms that, within endemic areas for CanL, it is also possible to observe clinical cases of EL caused by the same etiological agent causing disease in dogs, as has already been recorded in Spain and Portugal [15,16].

Even if *L. infantum* appears to be involved in most cases of EL reported in Europe [14,15,16], cases of EL attributed to *L. martiniquensis* (formerly *L. siamensis*), the causative agent of human VL and CL in Thailand and Myanmar, were described in Germany and Switzerland [20], as well as in the USA [21,23].

Moreover, given the frequent movements of horses across Mediterranean areas, the genetic identity of EL cases should always be confirmed, in horses as well as pets, due to the increasingly frequent reporting of cases determined by different species other than *L. infantum* (i.e., *Leishmania major* and *Leishmania tropica*) [36,37].

The positive sample (ID 18) showed the presence of nodular lymphoplasmacytic and histiocytic dermatitis, extending from the superficial to the deep dermis and composed of histiocytes and lymphocytes, with fewer epithelioid cells and multinucleated giant cells and lesser plasma cells.

Reports of *Leishmania*-associated skin lesions in horses described granulomatous dermatitis with macrophages, lymphocytes, multinucleated giant cells, fewer neutrophils and eosinophils, and variable amounts of intralesional amastigotes as a consistent pattern [20,38]. In some cases, lesions were more pyogranulomatous, with large numbers of neutrophils and eosinophils in the inflammatory infiltrate [23].

Neutrophils and eosinophils were sparse in our case, though more abundant in other horses in our case series, where the histologic lesions were suggestive of hypersensitivity diseases.

*Leishmania* amastigotes could be successfully detected in the cytoplasm of macrophages via immunohistochemistry, using a hyperimmune serum from a dog naturally infected with *L. infantum* [39] as the primary antibody. The cross-reactivity of the canine serum in the horse skin was expected, due to the 99–100% homology of the sequence amplified from the FFPE skin with the GenBank sequences of *L. infantum*. The use of a canine hyperimmune serum as the primary antibody to detect leishmanial amastigotes in equine skin has already been used [40] and proves to be an alternative and relatively inexpensive technique to demonstrate parasites in biopsies.

Worthy of note in our *Leishmania*-positive case is the finding of dermal lymphoid aggregates with germinal center formation. These structures, called tertiary follicles, can develop in non-lymphoid organs in response to chronic antigenic stimulation, inflammation, or persistent infection [41]. Dermatitis reactions due to different species of *Leishmania* spp. associated with T- and B-cell hyperplasia have been described in humans and diagnosed as cutaneous lymphoid hyperplasia or cutaneous pseudolymphoma [42,43,44]. Whether this pattern can be a manifestation of *Leishmania* dermatitis in horses remains to be confirmed.

The finding of a single positive horse through molecular investigations might support the marginal role of equines as reservoirs of infection, at least in Italy; in fact, the absence of parasites in most of the skin samples of exposed equine populations seems to indicate a scarce possibility for sand flies to acquire the infection through the blood meal of this species, differently from what happens even in the absence of clinical lesions in dogs [45].

The role of horses living close to anthropized contexts (e.g., farms or equestrian and recreational centers) in epizootic areas for *Leishmania* may still require clarification. However, the moderate seroprevalence values detected in apparently healthy equine populations [28,35,46], together with the opportunistic feeding behavior of most of the phlebotomine species (i.e., *Phlebotomus perniciosus, Phlebotomus perfiliewi, Phlebotomus papatasi*, and *Phlebotomus mascittii)* acting as vectors of *L. infantum* in the Mediterranean area [47], support the indirect role of horses in the epizoological scenario of CanL and ZVL [28,35,46].

## 4. Materials and Methods

### 4.1. Skin Sampling Collection

The database repositories of Veterinary Pathology Services, Veterinary Teaching Hospitals (OVUDs), and Departments of Veterinary Medicine of Perugia (Umbria region), Milan (Lombardy region), and Bologna (Emilia Romagna region) were searched to identify histopathological records of skin biopsies from horses living in CanL-endemic areas of north-central Italy, where exposure of equine populations to *Leishmania* infection was ascertained by previous serological investigations [30].

The skin FFPE blocks of the horses were included in the study if (i) the histopathology previously performed by a board-certified veterinary pathologist (C.B., P.R., and G.A.) confirmed features consistent with chronic dermatitis and an inflammatory infiltrate characterized by macrophages, lymphocytes, and plasma cells; (ii) an adequate amount of FFPE tissue (>0.5 cm^2^) was stored in the biorepository at the time of the investigation.

The present study included 47 horses (ID: 1–47), whose samples were collected between 2008 and 2021; of these, 25 were obtained from the repository of Perugia, 6 from Bologna, and 16 from Milan.

### 4.2. Histopathology and Immunohistochemistry

Slides containing 4 µm sections were prepared from the skin FFPE blocks and stained with H&E for histopathological examination. Hematoxylin-and-eosin-stained slides were assessed for lesions in the dermis as follows: severity (mild, moderate, severe); pattern (perivascular, interstitial, band-like, nodular, diffuse); and depth (superficial, mid, deep, or any combination) of dermal inflammation.

The biomolecular positive sample block was cut, and multiple slides were utilized for histochemical special stains, including periodic acid–Schiff (PAS), Grocott, and Ziehl–Neelsen, and submitted to immunohistochemistry (IHC) as described [39], using the serum of a dog naturally infected with *L. infantum* as the primary antibody. A known positive skin sample from a CanL-sick dog was used as a positive control. The negative control was incubated with Tris-buffered saline (TBS), omitting the primary antibody.

### 4.3. Molecular Methods

From the forty-seven samples, four-to-six 10 um-thick sections of each FFPE skin tissue block were cut and handled with a new disposable razor blade and new gloves to prevent cross-contamination with *Leishmania* DNA. After cutting each block, the microtome blade, tweezers, and entire cutting area were carefully cleaned with a 0.1 M solution of sodium hypochlorite to break down any potential contamination.

The sections were deparaffinized at room temperature by immersion washing twice for 30 min each in 1 mL of xylene and rinsed twice with 1 mL of 100% ethanol for 5 min. The samples were centrifuged at 10,000× *g* for 5 min, and the liquid was decanted between each change.

Total genomic DNA was extracted with the ExgeneTM FFPE Tissue DNA Kit (GeneAll, Seoul, Korea), according to the manufacturer’s protocol. Since formalin-fixation of histological specimens causes partial destruction of DNA, which may hamper diagnostic PCR analysis, and a broad spectrum of *Leishmania* species may infect horses, we decided to test the extracted DNA by using two conventional (c)-PCR protocols: amplifying a short fragment of 120 bp of the conserved region of the *Le**ishmania* kinetoplastic (k) DNA minicircle [48] and a 330 bp fragment of the *Internal transcribed spacer-1* (ITS-1) of the ss-rRNA [49] (Table 1).

The reactions were carried out in a StepOnePlus™ instrument (Applied Biosystems, Foster City, CA, USA). Each reaction included a negative (DNA-, nuclease-free water) and positive control, which consisted of DNA extracted from *L. infantum*-cultured promastigotes.

A total of 15 μL of the amplification products were submitted to 1.2% agar gel electrophoresis for 30 min at 100 V in TBE buffer (89 mM Tris borate, 2.0 mM EDTA, pH 8.3), using 5 uL of EuroSafe Nucleic Acid Stain (EuroClone, Milan, Italy) and 10 uL of SharpMassTM 100 ready-to-load DNA ladder (100 bp) (EuroClone, Milan, Italy) to determine the PCR fragment size. The gels were visualized under UV transillumination.

The amplified products obtained from PCRs were directly sequenced in both directions using a 16-capillary ABI PRISM 3130 × l Genetic Analyzer, assembled and edited with SeqScape software v 2.5 (Applied Biosystem, Foster City, CA, USA). The assembled sequence was compared to *Leishmania* spp. sequences available in GenBank using a Basic Local Alignment Search Tool (BLAST, https://blast.ncbi.nlm.nih.gov/, accessed on 20 December 2021) and then aligned with representative sequences using MegaX software [51]. 

Genomic DNA was also employed in a quantitative (q)PCR protocol, amplifying a fragment of the k-DNA minicircle [51]. For this tool, serial 10-fold dilutions of recombinant plasmid containing the target DNA with a known copy number (from 1 × 10^7^ to 1 × 10 copies/μL) were used to generate the qPCR standard curve. The qPCRs were performed in a 20 μL volume consisting of 2 μL of DNA sample (10 ng), 20 μM of each primer, 2 × TaqMan Master Mix (Applied Biosystems, Foster City, CA, USA), and 10 μM of fluorogenic probe5′ Fam-TGGGTGCAGAAATCCCGTTCA-3′ BHQ1 (Applied Biosystem, Foster City, CA, USA). The exogenous internal control was supplied by the TaqMan exogenous internal positive control reagents kit (Applied Biosystems, Foster City, CA, USA). Reactions were run on a 7500 ABI Prism sequence detection system (Applied Biosystems, Foster City, CA, USA). Each run included a positive control sample (DNA from cultured promastigotes of *Leishmania*), a negative control (DNA nuclease-free water), and a blank (no-template control). Each sample was tested in duplicate. 

## 5. Conclusions

The finding of this first positivity to *L. infantum* on a cutaneous lesion of a horse from a CanL-endemic area of Italy, although in fact apparently rare, is certainly noteworthy; therefore, this finding supports the inclusion of EL in the differential diagnosis of papular or nodular skin lesions in equine species.

## Figures and Tables

**Figure 1 pathogens-11-00634-f001:**
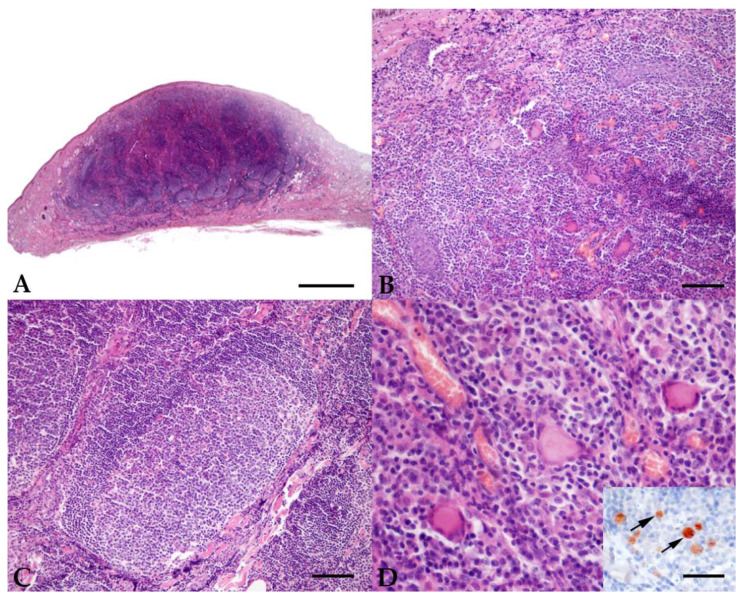
Case ID 18. Sella Italiano horse, female, 8.5 years old. Histopathological sections from the skin lesion: (**A**) the superficial, mid and deep dermis is occupied by a nodular infiltration of inflammatory cells (scale bar = 1000 um); (**B**) inflammatory cells consist of a mixture of macrophages and lymphocytes, with fewer epithelioid macrophages, multinucleated giant cells, and plasma cells (scale bar = 100 um); (**C**) multifocal lymphocytes are arranged in nodular, pseudofollicular structures with central germinal centers (scale bar = 100 um); (**D**) Langhans-type multinucleated giant cells are present with intracytoplasmic *Leishmania* amastigotes, as demonstrated by the immunohistochemical staining (scale bar = 50 um; the arrows point to the amastigotes in the cytoplasm of macrophages).

**Table 1 pathogens-11-00634-t001:** Oligonuclotides and thermal cycling protocols of c-PCRs and qPCR reactions.

Gene Target	Primers	Product (bp)	Annealing t°	Cycling Protocol	Cycle Numbers	Reference
**k-DNA**	F 5′-AACTTTTCTGGTCCTCCGGG-3′ R 5′-CCCCCAGTTTCCCGCCC-3′	120	58 °C	denaturation 94 °C for 60 s annealing 58 °C for 30 s extension 72 °C for 30 s	35	Francino et al., 2006 [48]
**ITS-1**	F 5′-CTGGATCATTTTCCGATG-3′ R 5′-TGATACCACTTATCGCACTT-3′	330	51 °C	denaturation 94 °C for 30 s annealing 51 °C for 45 s extension 72 °C for 60 s	35	El Tai et al., 2000 [49]
**k-DNA (qPCR)**	F 5′-GGCGTTCTGCGAAAACCG-3′ R 5′-AAAATGGCATTTTCGGGCC-3′ probe 5′ Fam-TGGGTGCAGAAATCCCGTTCA-3′ BHQ1	68	60 °C	denaturation 95 °C for 15 s annealing 60 °C for 60 s extension 72 °C for 30 s	40	Vitale et al., 2004 [50]

## Data Availability

All study data are presented in the article.

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
