# Peer review of "Detection of *Leishmania* spp. in Chronic Dermatitis: Retrospective Study in Exposed Horse Populations"

_pathogens, 2022, doi:10.3390/pathogens11060634_

Round 1
Reviewer 1 Report
Gazzonis and coworkers evaluate the frequency of Leishmania infection in biopses from cutaneous lesions of horses living in the region with prevalence of canine leishmaniasis in Italy. This study is important to increase the knowledge about the role of horse as potential host.
I recommend the major corrections.
Include the figure of PCR amplification of samples, showing a. positive sample and alignments in the BALST. Include the Scale bar in the image of fig 1 and arrow to point the amastigotes.
Have some information about the cases of CanL in the same local of infected horse?
Author Response
Gazzonis and coworkers evaluate the frequency of Leishmania infection in biopses from cutaneous lesions of horses living in the region with prevalence of canine leishmaniasis in Italy. This study is important to increase the knowledge about the role of horse as potential host.
I recommend the major corrections.
Include the figure of PCR amplification of samples, showing a. positive sample and alignments in the BALST.
The pictures of the PCR products as well as of the Blast alignments were not included in the first version of our paper since we thought it was redundant to insert them because the sequence of the isolate was deposited in gene bank and moreover the literature is already rich of this kind of pictures, however if the reviewer wishes we could insert them as supplementary files.
Include the Scale bar in the image of fig 1 and arrow to point the amastigotes.
The scale bar and the arrows to point amastigotes have been added in Figure 1 and the legend of the figure has been adapted accordingly
Have some information about the cases of CanL in the same local of infected horse?
Unfortunately we don’t have any specific environmental anamnestic information about the presence of infected dog/dogs co-housing with the positive horse because the present paper consisted in a retrospective work based on skin samples obtained from histologic repositories; however, as reported in the paper, the same geographical areas was previously explored in a cross sectional epidemiological survey and showed an endemic condition for CanL
Reviewer 2 Report
Comments:
- Leishmania infection in mostly asymptomatic therefore, may have low level of parasitemia which cannot be detected by the the PCR. Since the authors found only 1/47 PCR positive horse, did they check the serological status of these presumably Leishmania infected horses so as to determine the prevalence of Leishmania infection among this cohort?
- Have the authors checked the serological homology using canine Leishmania serum against the parasites antigens isolated from horse infections?
- Since the authors found only one PCR positive horse, is it possible that there may be higher prevalence of Leishmania infection among the horses in the areas where dogs are sero-positive?
Author Response
Leishmania infection in mostly asymptomatic therefore, may have low level of parasitemia which cannot be detected by the PCR. Since the authors found only 1/47 PCR positive horse, did they check the serological status of these presumably Leishmania infected horses so as to determine the prevalence of Leishmania infection among this cohort?
The cases analyzed in our study were sent to the histopathology labs without any previous suspect about Equine Leishmaniosis (no suspicion or preliminary tests recorded in the history sheet). It was a case series selected using exclusively morphologic (H&E) inclusion criteria, blind to the infection status of the animals, with the aim to investigate if Leishmania should be implicated in the pathogenesis of such lesions. Based on the premises explained above, it was not possible to have information on the seroreactivity of this cohort of animals. In a longitudinal epidemiological survey we could have additional information on the exposition; however, the presence and prevalence of exposition was investigated in a previous epidemiological survey from horse populations reared in the same geographical areas as reported in the paper.
Have the authors checked the serological homology using canine Leishmania serum against the parasites antigens isolated from horse infections?
We did not check for seropositivity as immunohistochemistry was performed, as morphological confirmation, after receiving the PCR results and therefore after knowing that the horse was infected with Leishmania infantum. At that point, we used the serum from a dog naturally infected with L. infantum in analogy to when using sera from laboratory animals (mouse or rabbit) directed against a common antigen, in this case L. infantum. This method is commonly used in immunohistochemistry and indeed, using serum from a dog on horse, we avoided problems deriving by the possible cross-reactivity of the secondary antibody (anti-dog) to endogenous horse IgG, which occur when using a primary antibody created in the same species as the animal being tested ("mouse-on-mouse" staining strategy). The same method (canine serum as primary antibody) was indeed used by the paper we cited in the text.
Since the authors found only one PCR positive horse, is it possible that there may be higher prevalence of Leishmania infection among the horses in the areas where dogs are sero-positive?
Thanks for the comment, what the reviewer says is absolutely true, but it is difficult to define since there is very scant information about the sensibility of biological targets for biomolecular tools from horses (i.e. blood, or blood marrow, spleen etc); since the EL cases are referred to as cutaneous forms, the skin represents the target that has been most investigated from the biomolecular point of view and we don’t have further biological targets to be used as reference. May be a longitudinal survey using both serological and qPCR methods on skin could be more resolutive to give a real estimation of the level of exposure.